# GPSR: Gradient-Prior-Based Network for Image Super-Resolution

Xiancheng Zhu [1], Detian Huang [1,*], Xiaorui Li [2], Danlin Cai [3] and Daxin Zhu [3]

1 College of Engineering, Huaqiao University, Quanzhou 362021, China
2 College of Fine Arts, Huaqiao University, Quanzhou 362021, China
3 School of Mathematics and Computer Science, Quanzhou Normal University, Quanzhou 362021, China
* Correspondence: huangdetian@hqu.edu.cn

**Abstract:** Recent deep learning has shown great potential in super-resolution (SR) tasks. However, most deep learning-based SR networks are optimized via pixel-level loss (i.e., L1, L2, and MSE), which forces the networks to output the average of all possible predictions, leading to blurred details. Especially in SR tasks with large scaling factors (i.e., ×4, ×8), the limitation is further aggravated. To alleviate this limitation, we propose a Gradient-Prior-based Super-Resolution network (GPSR). Specifically, a detail-preserving Gradient Guidance Strategy is proposed to fully exploit the gradient prior to guide the SR process from two aspects. On the one hand, an additional gradient branch is introduced into GPSR to provide the critical structural information. On the other hand, a compact gradient-guided loss is proposed to strengthen the constraints on the spatial structure and to prevent the blind restoration of high-frequency details. Moreover, two residual spatial attention adaptive aggregation modules are proposed and incorporated into the SR branch and the gradient branch, respectively, to fully exploit the crucial intermediate features to enhance the feature representation ability. Comprehensive experimental results demonstrate that the proposed GPSR outperforms state-of-the-art methods regarding both subjective visual quality and objective quantitative metrics in SR tasks with large scaling factors (i.e., ×4 and ×8).

**Keywords:** super-resolution; deep learning; gradient prior; feature representation; spatial attention

## 1. Introduction

As one of the most crucial tasks in computer vison, Single Image Super-Resolution (SISR) aims to reconstruct a latent high-resolution (HR) image with plentiful high-frequency details from a single available low-resolution (LR) image by learning a complex nonlinear mapping. SISR has drawn much attention due to its wide range of practical applications, such as satellite imaging [1,2], medical imaging [3,4], face recognition [5,6], and video surveillance [7,8]. In recent years, deep learning has attracted increasing attention in SR tasks due to its powerful feature representation ability.

Dong et al. [9] first applied deep learning to SISR task by proposing SRCNN, which consists of three convolutional layers and which exhibits remarkable performance over traditional SR methods. Since then, numerous works have focused on deeper network structures [10], more extensive connections [11], more efficient attention mechanisms [12], and more powerful non-local operations [13,14] to enhance the feature representation of CNN-based SR models. Lim et al. [10] proposed EDSR with a very deep structure by stacking massive optimized residual blocks. Impressive performance improvement brought about by EDSR reveals that network depth plays an important role in high-quality image SR. To explore more useful SR cues from different aspects, Zhang et al. [11] proposed RDN to aggregate hierarchical features at different network depths adaptively. Further, Niu et al. [12] proposed HAN to recalibrate the hierarchical features by modeling the interdependencies among layers. In addition, Mei et al. focused on non-local operations in

an SISR task and successfully proposed NLSN [13] and PA-EDSR [14], which fully utilize numerous highly similar textures within the image to synthesize the high-fidelity images.

However, most deep learning-based SR methods are optimized via pixel-level loss (i.e., L1, L2, and MSE), which guides the SR process by minimizing the pixel-to-pixel distance between the reconstructed images and the HR reference ones. Under such a constraint, the SR model tends to output the average of all possible predictions, resulting in blurred details of the reconstructed images. More seriously, the limitation is further aggravated in SR tasks with large scaling factors (i.e., ×4 and ×8). Therefore, it is important to improve the high-frequency detail reconstruction capability of the pixel-level loss-based SR models.

Considering that the gradient map of an image can reflect the spatial distribution of the high-frequency information, but also identify the local regions where the high-frequency components exist, we attempt to exploit the gradient map to guide the network to restore the appropriate edge and texture details at the suitable positions. To this end, we fully explore the potential of the gradient prior for the SR task and then propose a detail-preserving Gradient-Guided Strategy (GGS) to attain sharp edge and texture details, and to preserve the geometric invariance of the reconstructed images. Specifically, on the one hand, due to the strong correlation between the gradient information and high-frequency components, the gradient information can be transformed into the sharp edge and texture features. Thus, we introduce an additional gradient branch in the classical network with a single SR branch. The introduced gradient branch works as a feature selector that adaptively extracts structural features to provide a gradient prior for the SR branch. On the other hand, we propose a compact Gradient-Guided (GG) loss function to avoid blindly restoring high-frequency details by constraining the spatial structure of the reconstructed images. The proposed GGS facilitates the suppression of undesired geometric distortions while preventing over-smoothing or over-sharp recovery.

Furthermore, the benefit from multi-level contextual residual features, which are hierarchical with different receptive fields, can provide valuable cues for SR task from different perspectives. We propose two Residual Spatial Attention Adaptive Aggregation Modules (RS3AMs) to fully utilize the hierarchical features from the original LR images and embed them in the SR branch and the gradient branch, respectively. Specifically, on the one hand, considering that most SR methods [10,13,15] suffer from the hierarchical features being learned in a local way, which hinders the SR performance, we propose a novel Contextual Residual Fusion Structure (CRFS). Our CRFS removes over-dense connections and frequent concatenation operations from the dense block of RDN [11], and adaptively learns global hierarchical features in a holistic way, which promotes its feature representation ability. On the other hand, to enhance the feature representation of CRFS, we propose an efficient Large-Receptive-field-based Spatial Attention Module (LRSAM) to adaptively recalibrate the local spatial information of the feature map. To the best of our knowledge, large receptive fields facilitate the network to fully exploit the interdependence of local spatial information and significantly strengthen the sensitivity of the SR model toward critical spatial content. Finally, we combine LRSAM with CRFS to acquire two RS3AMs for effective feature extraction in the SR branch and the gradient branch, respectively. It is worth mentioning that GPSR achieves better SR performance with smaller parameters (8.18M) compared to most state-of-the-art SR methods, such as SAN [15] (15.9M), HAN [12] (16.07M), and NLSN [13] (46.52M). Overall, our contributions are summarized as follows.

- We propose a Gradient-Prior-based Super-Resolution network (GPSR) for images. The experimental results show that the proposed GPSR achieves superior performance against state-of-the-art methods in terms of subjective visual results and objective evaluation metrics.
- We propose a detail-preserving Gradient-Guided Strategy (GGS) to prompt the model to focus more attention on the most critical high-frequency components, and to suppress undesired geometric distortions as much as possible.

- We propose two Residual Spatial Attention Adaptive Aggregation Modules (RS3AMs) and further incorporate them in the SR branch and the gradient branch, respectively, to fully explore and utilize the intermediate features to enhance feature representation capability.

## 2. Related Work

### 2.1. Gradient Prior

Extensive studies have confirmed that using common priors, which describe natural image properties, such as sparsity [16,17], spatial smoothness [18,19], non-local similarity [13–15,20], and gradient prior [21–23], can effectively strengthen the high-frequency details of the reconstructed images. Among them, the gradient prior, as one of the most critical priors, is widely used in SR tasks since it is easy to extract and can provide more advantageous information to generate sharp edges and textures. Yang et al. [22] proposed the first recurrent network model with residual learning for SR, which introduced the gradient prior between the LR images and HR images into the SR process to enhance the detail-preserving ability. To address the problem of structural distortion in GAN-based SR methods, Ma et al. [21] proposed a structure-preserving Super-Resolution (SPSR), which introduces additional gradient prior to guide the generator to restore straight and sharp edges while suppressing undesired spatial distortions as much as possible. Considering that recovering diverse low-level image elements at a single stage is not an optimal strategy due to the different characteristics of various low-level components in natural images, Wei et al. [23] proposed the Component Divide-and-Conquer (CDC) model, which produces attention masks corresponding to smooth, edge, and corner regions by utilizing gradient prior, to acquire the corresponding structural components, and then merges these three structural components as the final output. The above studies [21–23] have revealed that gradient prior is able to effectively improve the high-frequency details of reconstructed images. However, for SR tasks with large scaling factors (i.e., ×4 and ×8), these methods are unable to predict sharp edge and texture details, and even produce unacceptable geometric distortions due to the inability to supplement multi-scale structural information.

### 2.2. Contextual Residual Feature

As the network depth increases, multi-level contextual residual features, which are hierarchical with different receptive fields, can benefit SR from different perspectives. However, numerous deep-learning-based SR methods, such as VDSR [24], EDSR [10], SAN [15], PA-EDSR [14], NLSN [13], etc., constitute a deep network by simply stacking a massive quantity of residual modules, but they do not fully utilize the complementary contextual residual features, limiting the SR performance. To improve the information flow, Huang et al. [25] proposed a densely connected convolutional network (DenseNet), which uses dense connectivity to achieve a direct connection between any two layers, significantly improving the efficiency of feature and gradient information transfer, and reducing the training difficulty. However, DenseNet is unsuitable for the SR task. On the one hand, the Batch-Normalization (BN) layers severely corrupt the contrast information of the reconstructed images. On the other hand, the pooling operation may lose crucial spatial information. To address this issue, Zhang et al. [11] designed a Residual Dense Block (RDB), which densely connects abundant local features to generate more advanced fused features while removing modules that negatively impact on the SR task. Nevertheless, over-dense connections and over-frequent concatenation operations not only require a large amount of computing resources, but they also introduce redundant feature information which hinders the feature representation. To reduce the heavy computation burden, Liu et al. [26] proposed a Residual Feature Aggregation framework (RFA). Compared with RDB, RFA removes unnecessary skip connections and directly forwards the features on each local residual branch to the end of the network, leading to a better SR performance. Additionally, inspired by the feedback mechanism that permits the network to carry a signal of output to correct previous states, Li et al. [27] propose an image Super-Resolution FeedBack Network

(SRFBN) to refine low-level representations with high-level information in a top-down manner to promote the SR performance. The above studies [11,25–27] have shown that the rational exploration and utilization of contextual residual features can provide more helpful information for the SR task. However, due to the lack of discriminative learning capability, a considerable number of redundant features may be passed to the end of the network and hinder the SR performance.

### 2.3. Attention-Based Networks

The visual attention mechanism is a visual information-processing mechanism unique to the human brain, which is able to acquire crucial local regions while suppressing irrelevant information by swiftly scanning the global image [28]. In the field of machine vision, the attention mechanism can be interpreted as an efficient way to manage resources by allocating more available computational resources to critical image signals. In recent years, numerous studies [12,15,29,30] have verified the effectiveness of attentional mechanisms. Considering that treating channel-wise features equally hinders the feature representation ability, Zhang et al. [29] proposed a very deep residual channel attention network (RCAN), which explores interdependencies among channels using the proposed Residual Channel Attention Module (RCAB) to adaptively reweight channel-wise features, thus prompting the network to focus on critical information. Dai et al. [15] proposed a Second-Order Attention Network (SAN), which endeavors second-order feature statistics to fully explore the dependencies of intermediate features to improve the feature representation ability further. Mei et al. [30] proposed the first Cross-Scale Non-Local (CS-NL) attention module, which combines the new CS-NL prior with local and in-scale non-local priors to reconstruct high-quality images. Considering that most attention mechanisms act only on each independent network layer and they ignore the relevance of intermediate layer features, Niu et al. [12] proposed the holistic attention network (HAN), which models the interdependencies between intermediate layers and adaptively highlights significant multi-level features. The above studies [12,15,29,30] have confirmed that the attention mechanism is able to adaptively highlight crucial features and effectively enhance the feature representation ability. However, existing spatial attention modules [31,32] only capture spatial information that are beneficial for SR tasks in extremely localized ways due to the lack of a large receptive field. Although the non-local operations [13–15,30,33] provide more global information to the network by capturing long-range dependencies, their high computational cost is not negligible.

### 3. Method

Most SR methods suffer from blindly recovering high-frequency details, resulting in their reconstructed images lacking sharp edge and texture details, along with undesired geometric distortion and artifical artifacts [34,35]. Particularly, this problem becomes more severe as the scaling factor increases. To address the problem, we propose a Gradient-Prior-based Super-Resolution network (GPSR), which is able to reconstruct sharp edge and texture details in SR tasks with large scaling factors (i.e., $\times 4$ and $\times 8$). In this section, we first introduce the overall architecture of our GPSR, followed by a detailed description of the detail-preserving Gradient-Guided Strategy (GGS) and the Residual Spatial Attention Adaptive Aggregation Module (RS3AM).

### 3.1. Network Architecture

As shown in Figure 1, our GPSR is composed of two branches, including the gradient branch and the SR branch. Among them, the gradient branch contains several gradient modules, which are mainly responsible for inferring multi-scale spatial structural feature maps to provide additional gradient priors for each decoding stage of the SR branch, thus improving the edge and texture details of the reconstructed images; while the SR branch contains several encoder–decoders, and it aims to generate the reconstructed images with high fidelity.

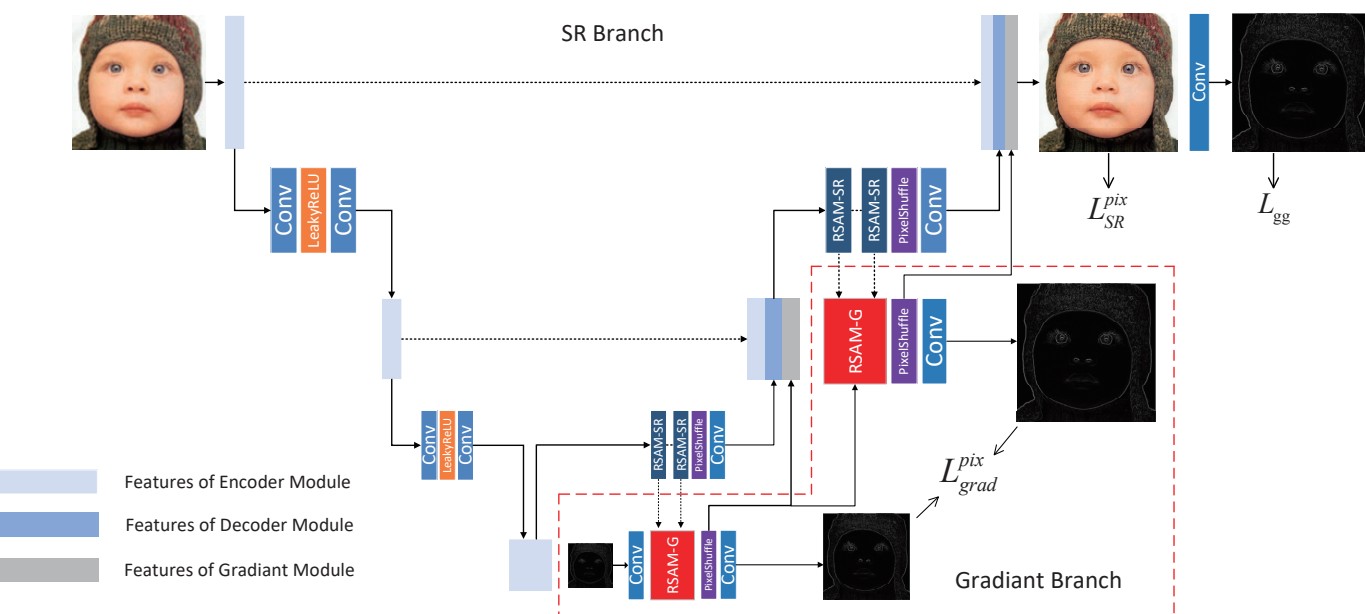

**Figure 1.** Network architecture of our Gradient-Prior-based Super-Resolution network (GPSR). The network consists of two branches, the gradient branch (bottom right), which is used to generate the multi-scale spatial structure feature maps, and the SR branch (upper left), which is used to generate the final reconstructed image. Notably, the intermediate features generated on the SR branch are integrated into the gradient branch to provide additional beneficial structural features. In addition, the multi-scale spatial structure feature maps generated on the gradient branch are integrated into the SR branch to strengthen the edge and texture details.

Given a LR image $I_{LR}$ as the input of our GPSR, the bicubic upsampling is first used to rescale $I_{LR}$ to the desired resolution, and then the shallow feature $F_0$ is extracted from $I_{LR}$ through a convolutional layer (Conv),

$$F_0 = f_{conv}(f_{up}(I_{LR})), \tag{1}$$

where $f_{up}(\cdot)$ denotes the upsampling function with bicubic kernel and $f_{conv}(\cdot)$ denotes the convolution operation.

The shallow feature map $F_0$ passes through $N$ stacked encoders, yielding $N$ shallow feature maps with different scales additionally. Let $F_{i-1}$ be the input of the $i$-th encoder; the corresponding output $F_i$ can be obtained by:

$$F_i = H_{en}(F_{i-1}), \tag{2}$$

where $H_{en}(\cdot)$ denotes the encoder function and $i = 1, 2, ..., N$. The goal of the encoder, which contains two stacked Convs (with stride 2) and a Rectified Linear Unit (ReLU), is to perform $\times 2$ downsampling (halving the spatial dimension and doubling the channel dimension) and shallow feature extraction operations:

$$H_{en}(\cdot) = f_{conv}(\rho(f_{sconv}(\cdot))), \tag{3}$$

where $f_{sconv}(\cdot)$ represents the function of the strided Conv (with stride 2), and $\rho(\cdot)$ represents Leaky ReLU.

To compensate for the information loss caused by forward propagation, we pass these shallow features $F_i$ to the decoders at each level through skip connections, where $i = 0, 1, \cdots, N-1$.

Then, the output $F_N$ of the $N$-th encoder passes through $N$ decoders, yielding $N$ deep feature maps with different scales. Let $F_{j+N-1}$ be the input of the $j$-th decoder; the output $F_j'$ of the $j$-th encoder can be obtained by:

$$F_j' = H_{de}(F_{j+N-1}), \tag{4}$$

where $H_{de}(\cdot)$ denotes the decoder function, $j = 1, 2, ..., N$. The goal of the decoder, which contains $M$ stacked RS3AM-SR (Section 3.3.3) and a pixel-shuffle [36] layer followed by a Conv, is to perform $\times 2$ upsampling (halving the channel dimension and doubling the spatial dimension) and deep feature extraction operations:

$$H_{de}(\cdot) = f_{conv}\left(f_{subpix}\left(f_{de}^M\left(...f_{de}^2\left(f_{de}^1(\cdot)\right)...\right)\right)\right), \tag{5}$$

where, $f_{subpix}(\cdot)$ represents the sub-pixel convolution operation, $f_{de}^a(\cdot)$ represents the function of the $a$-th RS3AM-SR, and $a = 1, 2, ..., M$. Subsequently, by concatenating the shallow feature $F_{N-j}$, the output $F_j'$ of the $j$-th decoder and the spatial structure feature map $F_j^{grad}$ produced by the $j$-th gradient module, the advanced fused feature $F_{j+N}$ is acquired,

$$F_{j+N} = \left[F_{N-j}, F_j', F_j^{grad}\right]. \tag{6}$$

Then, $F_{j+N}$ is treated as the input of the $(j+1)$-th decoder.

Finally, the reconstructed image $I_{SR}$ is generated with a Conv:

$$I_{SR} = f_{conv}(F_{2N}). \tag{7}$$

where $F_{2N}$ denotes the final informative feature obtained through a series of decoders.

### 3.2. Detail-Preserving Gradient-Guided Strategy (GGS)

Considering that the gradient magnitude is able to effectively reflect the frequency of the image signal, in other words, the gradient magnitude is higher in the edge and texture regions corresponding to the high-frequency regions, and lower in the smooth regions corresponding to the low-frequency regions, the gradient information is able to highlight the high-frequency components beneficial for the SR task and filter out redundant low-frequency components. Consequently, a detail-preserving Gradient-Guided Strategy (GGS) is proposed to take advantage of the gradient prior from the LR images. Specifically, we design a novel gradient branch to adaptively strengthen high-frequency details and to provide gradient prior for the SR task. At the same time, we propose a compact Gradient-Guided (GG) loss that leads GPSR to infer appropriate gradient information at the suitable positions by learning the gradient space to avoid blindly recovering details.

#### 3.2.1. Gradient Branch

In the GPSR model, the additional gradient branch is designed to achieve the spatial distribution translation from the LR gradient images to the HR gradient images. Since the gradient map of an image can be acquired by calculating the difference among neighboring pixels, we perform convolution operations with specific kernels on an LR image $I_{LR}$ to attain its gradient map $I_{LR}^{grad}$:

$$I_{LR}^{grad} = g(I_{LR}) = \left\| \left[I_{LR} * M_x, I_{LR} * M_y\right]\right\|_2, \tag{8}$$

where $g(\cdot)$, $*$ and $\|\cdot\|_2$ denote the gradient feature extraction function, the convolution operation, and the $\ell_2$ norm, respectively. $M_x$ and $M_y$ separately represent the fixed convolution kernels for computing the horizontal and vertical gradients:

$$M_x = \begin{pmatrix} 0 & 0 & 0 \\ 1 & 0 & -1 \\ 0 & 0 & 0 \end{pmatrix}, M_y = \begin{pmatrix} 0 & 1 & 0 \\ 0 & 0 & 0 \\ 0 & -1 & 0 \end{pmatrix}. \tag{9}$$

As we know, the gradient map of an image contains a wealth of high-frequency information that can be converted into sharp edge and texture details. Therefore, the gradient branch is used to restore a series of multi-scale spatial structure feature maps through learning the complex mapping relationship between the gradient maps of the LR image and the corresponding gradient maps of the HR image. All spatial structure feature maps are delivered to each decoder in the SR branch, as depicted in Equations (4)–(6), to provide multi-scale gradient priors for the SR process.

Given an LR image $I_{LR}$ as the input of the gradient branch, we firstly convert $I_{LR}$ into the gradient map fo the LR image by utilizing the gradient extraction function $g(\cdot)$, and then we extract its shallow gradient feature $F_0^{grad}$ with convolution operation:

$$F_0^{grad} = f_{conv}(g(I_{LR})). \tag{10}$$

To restore beneficial spatial structure feature maps, it is common practice to stack a large number of basic blocks to form a very deep neural network for SR [35]. However, such an approach severely increases the number of parameters and leads to a bloated network. Considering that the intermediate features generated on the SR branch involve abundant high-frequency information, which are significant for structure preservation, we integrate these features into the gradient branch to recover more spatial structure feature maps with fewer parameters and better reconstruction accuracy. As shown in Figure 2, the output $F_{k-1}^{grad}$ of the $(k-1)$-th gradient module and the intermediate features $F_{de,k}^m$ produced by the $m$-th RS3AM-SR module in the $k$-th decoder on the SR branch are fed into the $k$-th gradient module to attain the spatial structure feature map $F_k^{grad}$:

$$F_k^{grad} = f_{conv}\left(f_{subpix}\left(H_{GE}\left(F_{k-1}^{grad}, F_{de,k}^1, ..., F_{de,k}^M\right)\right)\right), \tag{11}$$

where $H_{GE}(\cdot)$ denotes the function of RS3AM-G module (Section 3.3.3), and $F_{de,k}^m$ denotes the output of the $m$-th RS3AM-SR of the $k$-th decoder in SR branch, $m = 1, 2, ..., M$, $k = 1, 2, ..., N$. Then, $F_k^{grad}$ is transferred to the encoder instantly, supplementing the SR process with the essential structural information, as depicted in Equations (4)–(6). The proposed gradient branch contains $N$ gradient modules, which generate $N$ spatial structure feature maps at different scales to provide multi-scale gradient priors for restoring more sharp edge and texture details.

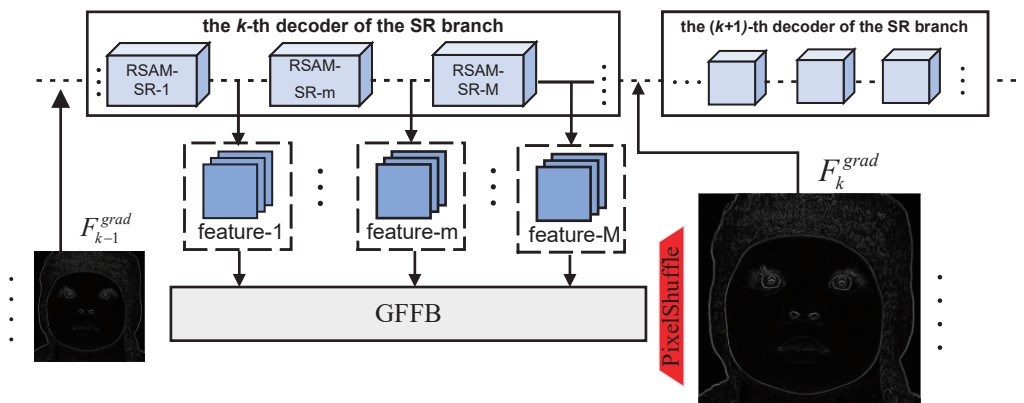

**Figure 2.** Architecture of the $k$-th gradient module of the gradient branch.

### 3.2.2. Gradient-Guided (GG) Loss

To avoid blindly restoring high-frequency details, we propose a compact GG loss that drives the network to generate appropriate gradient information at the suitable position. The proposed GG loss includes two items that affect the gradient branch and the SR branch, respectively.

On the one hand, inspired by ProSR [37] and LapSRN [38], we impose the deep multi-scale supervision strategy on the proposed gradient branch to provide accurate spatial structure information for the SR process. Specifically, the spatial structure feature map $F^s_{grad}$ generated by the $s$-th gradient module passes through a Conv, yielding the corresponding gradient map $G_s$:

$$G_s = f_{conv}\left(F^s_{grad}\right), \tag{12}$$

where $s = 1, 2, \cdots, N$. Each $G_s$ has its own loss and corresponding label $G'_s$, which is obtained via downsampling and gradient feature extraction operations on HR images $I_{HR}$:

$$G'_s = g(f_{down}(I_{HR})), \tag{13}$$

where $f_{down}(\cdot)$ represents the downsampling function with bicubic kernel. Since each $G_s$ and corresponding $G'_s$ have the same shape, the gradient branch can be optimized by $L^{pix}_{SR}$:

$$L^{pix}_{grad} = \sum_{s=1}^{N} \mathbb{E}\|G_s - G'_s\|_1, \tag{14}$$

where $\|\cdot\|_1$ denotes the $\ell_1$ norm.

On the other hand, we impose a gradient restriction on the reconstructed images, which enables the network to intensively learn the image gradient space. Such a scheme cannot only enhance the structure preservation capacity, but also alleviate over-smoothing or over-sharpening restoration. Specifically, we restrain the reconstructed image $I_{SR}$ by minimizing the distance of the gradient map between the HR image $I_{HR}$ and the reconstructed image $I_{SR}$,

$$L^{pix}_{SG} = \mathbb{E}\|g(I_{SR}) - g(I_{HR})\|_1. \tag{15}$$

Then, a compact GG loss $L_{gg}$ is formed by combining $L^{pix}_{grad}$ and $L^{pix}_{SG}$:

$$L_{gg} = \alpha L^{pix}_{grad} + \beta L^{pix}_{SG}, \tag{16}$$

where $\alpha$ and $\beta$ are the parameters to balance the $L^{pix}_{grad}$ item and the $L^{pix}_{SG}$ item, respectively. Empirically, $\alpha$ and $\beta$ were set to 0.1 and 0.001, respectively. Consequently, an overall loss $L_{GP}$ is proposed by combining pixel-wise and gradient-wise errors:

$$L_{GP} = L^{pix}_{SR} + L_{gg}, \tag{17}$$

where $L^{pix}_{SR}$ is the $L_1$ loss that minimizes the pixel-wise distance between the HR image $I_{HR}$ and its reconstructed image $I_{SR}$, and can be formulated as:

$$L^{pix}_{SR} = \mathbb{E}\|I_{SR} - I_{HR}\|_1. \tag{18}$$

Ultimately, the proposed GPSR is able to effectively recover gradient information and produce the reconstructed images with sharp edges and textures by minimizing $L_{GP}$.

### 3.3. Residual Spatial Attention Adaptive Aggregation Module

The fact that the hierarchical residual features available along the network depth are able to guide the SR process from different perspectives is ignored by most SR methods [10,11,13]. To address this issue, we propose a novel Contextual Residual Fusion

Structure (CRFS) to fuse complementary contextual residual features. Further, to enhance the feature learning capability of the CRFS, we propose an efficient Large-Receptive-field-based Spatial Attention Module (LRSAM) to adaptively recalibrate the spatial feature responses toward the most crucial of the input. The large receptive field is required for the attention block to work well for exploiting the interdependence of local information. Finally, we propose two residual spatial attention adaptive aggregation modules (i.e., RS3AM-G and RS3AM-SR), consisting of CRFS and a set of LRSAMs, for powerful feature extraction in the gradient branch and the SR branch, respectively.

### 3.3.1. Contextual Residual Fusion Structure (CRFS)

To take full advantage of the complementary contextual residual features, we propose a novel Contextual Residual Fusion Structure (CRFS). Different to the conventional single-path feed-forward framework, we redesign the arrangement of the stacked residual blocks [10], and then we add three residual blocks on the residual branch of CRFS to adaptively preserve the hierarchical features in a global way, as shown in Figure 3. Specifically, all contextual residual features are passed to a gating unit, which concatenates these multi-level features and adaptively learns the weights of different residual features through convolution operations. Let $R_{i,d}$ denote the $i$-th residual feature of the $d$-th CRFS, where $i = 1, 2, 3, 4$, then the corresponding fused feature $F_{con,d}$ is computed by:

$$F_{con,d} = H_{gate}([R_{1,d}, R_{2,d}, R_{3,d}, R_{4,d}]) \tag{19}$$

where $H_{gate}(\cdot)$ denotes the function of the gate unit. To improve the information flow and to compensate for the information loss during forward propagation, global residual learning is then employed to produce the output $F_d$ of the $d$-th CRFS:

$$F_d = F_{d-1} + F_{con,d} \tag{20}$$

where $F_{d-1}$ denotes the output of the $(d - 1)$-th CRFS.

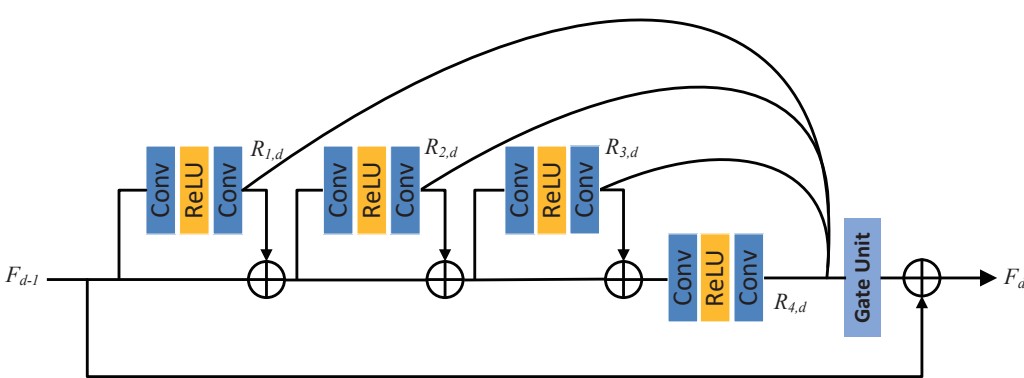

**Figure 3.** Architecture of the Contextual Residual Fusion Structure (CRFS).

### 3.3.2. Large-Receptive-Field-Based Spatial Attention Module

To promote the feature learning capability of CRFS, introducing a spatial attention mechanism is a reasonable solution. However, some spatial attention modules [32,33] are unable to effectively capture spatial location information that is helpful for SR task, due to the lack of large receptive fields. Non-local operations [13–15,30,33] serves as a promising solution to implement a spatial attention mechanism that explores correlations between any positions in a global way by capturing long-range dependencies. However, the large computational overhead is an insurmountable drawback. Considering that large receptive fields are able to fully exploit the interdependencies of local positional information, we propose an efficient Large-Receptive-field-based Spatial Attention Module (LRSAM) to enhance the sensitivity of the SR model to critical spatial content. Compared to non-local

operations, LRSAM has a lower computational complexity and can be easily embedded in different network positions.

LRSAM consists of two sub-modules, including the squeeze module and the excitation module, as shown in Figure 4. The squeeze module starts with a Conv to reduce the channel number of the input feature $Y_0$, then yielding a large receptive field by a stride Conv (with stride 3) and the max-pooling operation with a large window (e.g., $7 \times 7$) for enhancing the sensitivity of critical spatial content. To generate the spatial attention map, the squeeze module firstly applies a set of Convs followed by ReLU on the output $Y_1$ of the squeeze module. In the excitation module, $Y_1$ is passed through an upsampling layer of pixel-shuffle [36] and sigmoid activation to produce a spatial attention map $Y_{mask}$, and then $Y_{mask}$ is used to rescale $Y_0$ for highlighting beneficial feature information and to obtain $Y_0^*$.

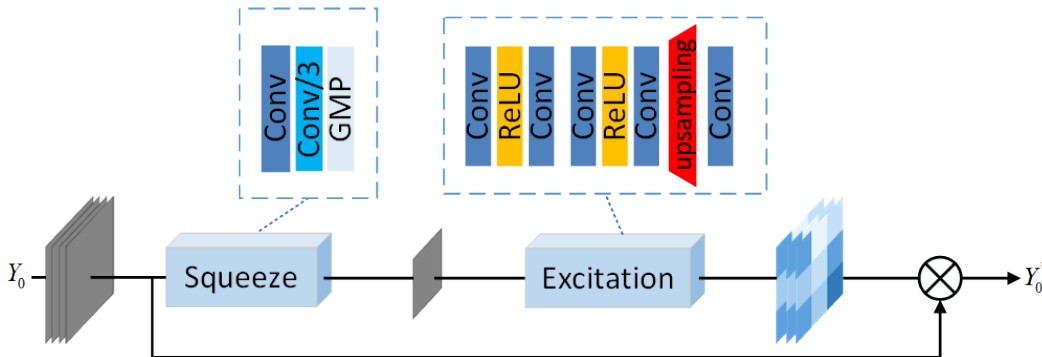

**Figure 4.** Architecture of the Large-Receptive-field-based Spatial Attention Module (LRSAM).

### 3.3.3. RS3AM-SR and RS3AM-G

To further improve the performance of CRFS, LRSAM is incorporated into the CRFS to construct two Residual Spatial Attention Adaptive Aggregation Modules (i.e., RS3AM-SR, RS3AM-G), which are embedded in the SR branch and the gradient branch, respectively, for efficient feature extraction. RS3AM-SR and RS3AM-G differ in their structural designs, depending on the characteristics of the different branches.

The architecture of RS3AM-SR that acts on the SR branch is depicted in Figure 5a. We embed a series of LRSAMs at the tails of residual branches at each level of CRFS. These LRSAMs are able to highlight spatial features that are useful for the SR task, and these highlighted features are passed through the gate unit to produce more beneficial features. Specifically, let $R_{c,d}^{sr}$ denotes the $c$-th residual feature of the $d$-th RS3AM-SR, which is enhanced by LRSAMs, where $c = 1, 2, ..., M$, then the corresponding fused feature $F_{con,d}^{sr}$ is computed by:

$$F_{con,d}^{sr} = H_{gate}([R_{1,d}, R_{2,d}, ..., R_{M,d}]). \tag{21}$$

To compensate for the information loss during forward propagation and to improve the information flow, global residual learning is then employed to obtain output $F_d^{sr}$ of the $d$-th RS3AM-SR:

$$F_d^{sr} = F_{con,d}^{sr} + x, \tag{22}$$

where $x$ denotes the input of the $d$-th RS3AM-SR.

The architecture of RS3AM-G that acts on the gradient branch is depicted in Figure 5b. As the intermediate features of the SR branch carry a wealth of high-frequency information, RS3AM-G utilizes them as an essential complement to inferring sharp spatial structure features. Specifically, the feature map that is concatenated by the output $F_{de,k}^m$ of the $m$-th RS3AM-SR of the $k$-th decoder and the local residual feature $F_{m,k}^{local}$ on $m$-th residual branch of the $k$-th RS3AM-G passes through an LRSAM and a Conv, yielding a fused feature $R_{m,k}^{grad}$:

$$R_{m,k}^{grad} = f_{conv}\left(f_{LRSA}\left(\left[F_{de,k}^m, F_{m,k}^{local}\right]\right)\right), \tag{23}$$

where $f_{LRSA}$ denotes the function of the LRSAM, $m = 1, 2, ..., M$, $k = 1, 2, ..., N$. It should be noted that the function of Conv here is to halve the channel dimension.

Subsequently, local feature fusion [39] is applied to adaptively fuse all intermediate features $R_{m,k}^{grad}$ in the current RS3AM-G, and then local residual learning is introduced in RS3AM-G to introduce the input $x$ of the $k$-th RS3AM-G to further improve the information flow. Thus, the output $F_k^G$ of the $k$-th RS3AM-G can be obtained by:

$$F_k^G = H_{gate}\left( \left[ R_{1,k}^{grad}, R_{2,k}^{grad}, ..., R_{M,k}^{grad} \right] \right) + x. \tag{24}$$

In contrast to dense block [25] and its variants [11], our RS3AM removes modules that have negative impacts on the SR performance, such as batch normalization (BN) layers, while simplifying excessive dense connections and frequent concatenating operations, and reinforces the spatial distribution of residual features at all levels by using LRSAM. Hence, the proposed RS3AMs are sufficiently efficient and lightweight. Finally, we used both RS3AM-SR and RS3AM-G to construct GPSR. Compared to most state-of-the-art SR methods, such as SAN (15.9M), HAN (16.07M), PA-EDSR (45.53M), NLSN (46.52M), etc., GPSR achieves a better super-resolution with fewer parameters (8.18M).

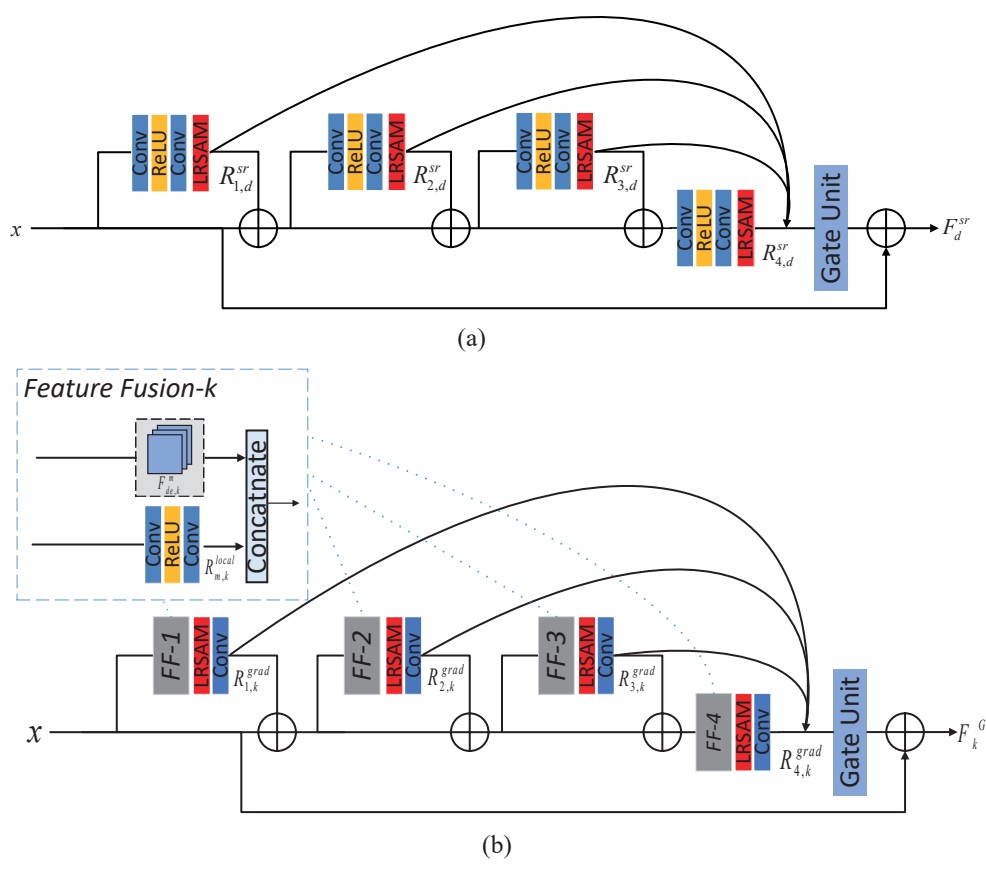

**Figure 5.** Two kinds of residual space attention adaptive aggregation module architectures (i.e., RS3AM-SR and RS3AM-G). (**a**) Architecture of the d-th RS3AM-SR that acts on the SR branch. (**b**) Architecture of the k-th RS3AM-G that acts on the gradient branch. For convenience, we only show the architectures of these two models separately when $M = 4$.

## 4. Experiments

In this section, we first present the experimental settings regarding datasets, degradation models, quantitative metrics, and training settings, then we visually validate the effectiveness of the Gradient-Guided Strategy (GGS), and comprehensively analyze the

contributions of the proposed three modules, including Large-Receptive-field-based Spatial Attention Module (LRSAM), Contextual Residual Fusion Structure (CRFS), and GGS. Finally, we compare the proposed Gradient-Prior-based Super-Resolution network (GPSR) with the state-of-the-art methods in terms of both quantitative metrics and visual quality.

*4.1. Settings*

4.1.1. Datasets and Degradation Models

Following [29,40], DIV2K [41] were selected as training sets to train the proposed GPSR, while five commonly benchmark datasets were used for testing, including set5 [42], set14 [43], BSD100 [44], Urban100 [45], and Manga109. Table 1 reports the characteristics of each dataset. Our experiments are separately conducted with bicubic (BI) and blur-downscale (BD) degradation models. In each training batch, 8 LR color patches with the size of $48 \times 48$ are extracted as the input of our GPSR.

**Table 1.** Characteristics of the dataset used for experiments.

| Dataset | Number of Images | Size | Main Content | Function |
|---|---|---|---|---|
| DIV2K [41] | 900 | 5.8 GB | Humans, animals, and landscapes | Training |
| Set5 [42] | 5 | 1.30 MB | Humans and animals | Testing |
| Set14 [43] | 14 | 7.44 MB | Humans and animals | Testing |
| BSD100 [44] | 100 | 38.87 MB | Landscapes and animals | Testing |
| Urban100 [45] | 100 | 194 MB | Buildings | Testing |
| Manga109 | 109 | 219 MB | Manga | Testing |

4.1.2. Quantitative Metrics

To evaluate the quality of the reconstructed images, we first convert the obtained reconstructed images into YCbCr space and then calculate the corresponding Peak signal-to-noise ratio (PSNR) and structural similarity (SSIM) metrics on the Y channel. This is because it is more efficient to calculate the metrics on the Y channel of the converted YCbCr space than on the three channels of the original RGB color space. It is worth noting that the high metric means a better super-resolution performance.

4.1.3. Training Settings

For training, we apply the Adam optimizer [46] with $\beta_1 = 0.9$, $\beta_2 = 0.999$, $\varepsilon = 10^{-8}$. This is because the Adam optimizer has the advantages of fewer hyperparameters, efficient computation, and fast convergence. The learning rate is initialized to $10^{-4}$ and decreased to $10^{-7}$ with a cosine annealing out of $10^6$ iterations in total. Furthermore, in an experiment with scaling factor $S$ (i.e., $\times 4$, $\times 8$), the numbers of encoders, decoders, and gradient modules are set to $N = \log_2^S$; the number of RS3AM-SR and the number of residual branch of RS3AM-G are set to $M = 4$. Our GPSR has been executed on the PyTorch framework and on an Nvidia GeForce RTX 3090 24GB GPU.

*4.2. Ablation Experiments*

We analyze the effectiveness of the proposed modules, including GGS, LRSAM, and CRFS, via ablation experiments. The baseline model is built upon the proposed GPSR by removing the GB and replacing the RS3AM-SR modules, which consist of CRFS and LRSAM, of the SR branch with the plain residual modules [10].

**Analysis on the gradient branch:** To verify the effectiveness of the GB, we visualize the output of the GB. Figure 6 illustrates the visual results of the GB. As can be seen from Figure 6a,d, the gradient maps of the HR images, which not only contain the sharp edge, but also effectively reflect the spatial structure relationship, are obtained by performing gradient feature extraction on the original HR images. This motivates us to design a detail-preserving GB to generates the gradient maps of the reconstructed images containing sharp geometric structures, and provides the essential gradient guidance for the SR process. Specifically, through the introduced GB, the gradient maps of the LR images (shown in

Figure 6b,e) were converted to the gradient maps of the reconstructed images (shown in Figure 6c,f). It can be observed that the gradient maps of the LR images are blurred and smooth in general, while the spatial structure details contained in the gradient maps of the reconstructed images (shown in Figure 6c,f) are significantly sharper and highly similar to the gradient maps of the HR images (shown in Figure 6a,d). This experimental result shows that the proposed GB possesses the ability to restore high-fidelity edge and texture information. This is because the proposed GB is able to generate the gradient maps at different scales and simultaneously transmit them as gradient priors for each upsampling stage on the SR branch to complement the multi-scale spatial structure features.

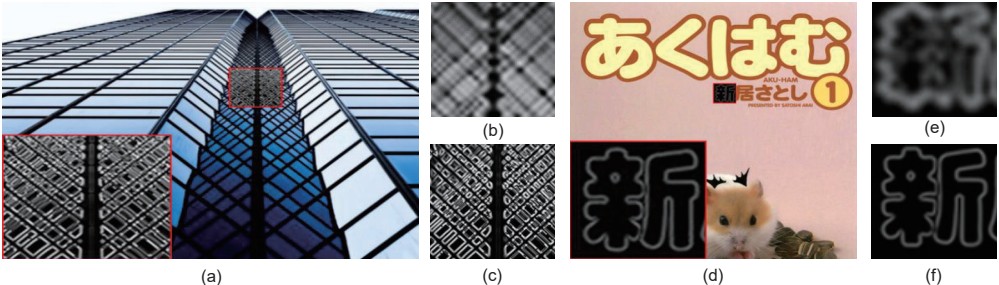

**Figure 6.** Results of gradient branch. Among them, (**a**,**d**) are the gradient maps of the reference HR images, (**b**,**e**) are the gradient maps of the corresponding LR images, and (**c**,**f**) are the gradient maps of the reconstructed images generated by the gradient branch.

**Analysis on gradient-guided loss:** We analyze the effectiveness of gradient-guided (GG) loss by removing $L_{grad}^{pix}$ or $L_{SG}^{pix}$ from the baseline model with the GGS, and report the results in Table 2. Here, $L_{grad}^{pix}$ is used to supervise the outputs of the GB to ensure the accuracy of the multi-scale gradient maps. Additionally, $L_{SG}^{pix}$ is used to supervise the gradient map of the final SR image. When $L_{grad}^{pix}$ is removed, the GB of the corresponding model cannot be optimized, and it outputs unreliable multi-scale gradient maps, resulting in SR performance degrading from 32.61 dB to 32.31 dB. When $L_{SG}^{pix}$ is removed, the SR performance of the corresponding model is reduced from 32.61 dB to 32.54 dB. It is because $L_{grad}^{pix}$ can effectively avoid the SR model from blindly restoring high-frequency information. From the above analysis, optimizing $L_{grad}^{pix}$ and $L_{SG}^{pix}$ jointly is essential. Consequently, the proposed gradient-guided loss can guide the network to reconstruct accurate high-frequency details by taking full advantage of $L_{grad}^{pix}$ and $L_{SG}^{pix}$.

**Table 2.** Ablation experiments on gradient-guided loss, including $L_{grad}^{pix}$ and $L_{SG}^{pix}$. The average PSNR values are evaluated on Set5 dataset with scaling factor $\times 4$.

| $L_{grad}^{pix}$ | ✓ | | ✓ |
|---|---|---|---|
| $L_{SG}^{pix}$ | | ✓ | ✓ |
| PSNR | 32.54 | 32.31 | 32.61 |

**Analysis on gradient guidance strategy:** We visually compared the baseline with and without the GGS, including the GB and the GG loss. Specifically, the LR images (shown in Figure 7a,d) as the network input, the reconstructed images, and the corresponding absolute gradient intensity maps (shown in Figure 7b,e) obtained using the model without GGS, and the reconstructed images and the corresponding absolute gradient intensity maps (shown in Figure 7c,f) obtained using the model with GGS, are depicted in Figure 7. Notably, the absolute gradient intensity maps directly reflect the quality of edge and texture details of the reconstructed images by using the brightness of color. The brighter the color

is, the sharper the structure details are. From Figure 7b,c, it can be seen that the sharpness of the whiskers in the reconstructed image is significantly improved by adopting GGS. Additionally, compared with Figure 7e, the wire on the window in Figure 7f appears sharper, which is more evident in the absolute gradient intensity map in Figure 7f. Additionally, the PSNR of the reconstructed images (i.e., Figure 7c,f) obtained by the model with GGS is significantly improved by 0.09 dB and 0.38 dB, respectively, compared to the baseline. It indicates that the proposed GGS not only strengthens the edge and texture details, but also alleviates the geometric distortion well.

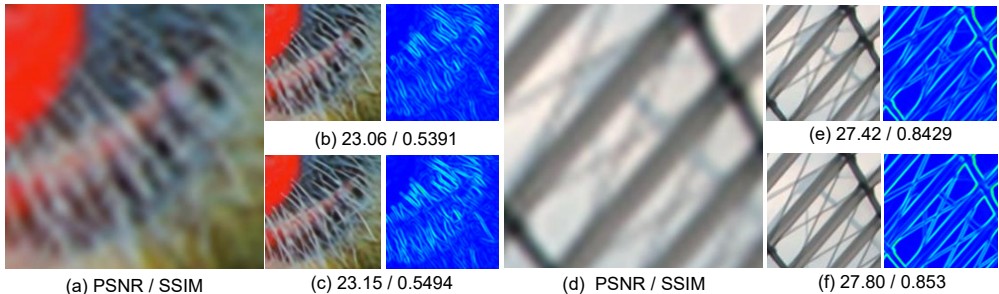

**Figure 7.** Visual comparison of the baseline with and without the Gradient-Guided Strategy (GGS). We present the LR images, the reconstructed images, and the absolute gradient intensity maps corresponding to the reconstructed images. Among them, (**a**,**d**) are input LR images, (**b**,**e**) are reconstructed results of baseline without GGS, and (**c**,**f**) are reconstructed results of baseline with GGS. The proposed GGS is able to effectively preserve the structure and acquire sharper high-frequency details.

**Analysis on effects of CRFS, CRFS, and GGS:** We conduct ablation experiments to analyze the effectiveness of CRFS, CRFS, GGS and their combinations, and report the results in Table 3. Compared with the baseline, the models with only CRFS or LRSAM achieve better results by up to 0.06 dB and 0.06 dB in terms of the PSNR metric, respectively. These results demonstrate that fusing the contextual residual feature can provide more advanced features for SR task, and spatial attention mechanisms with large-sized receptive fields can effectively explore valuable spatial content. At the same time, compared with the baseline, the PSNR obtained by the model with only GGS improves by 0.15 dB. It can be seen that our GGS possesses the highest contribution to performance improvement. This is because GGS provides the beneficial gradient prior for the SR process, which facilitates the guidance of the GPSR model to restore sharp high-frequency details. Furthermore, the improvement of the model with both CRFS and LRSAM is significant, with a PSNR improvement of 0.17 dB over the one with only CRFS, which indicates that LRSAM is able to effectively promote the feature representation ability of CRFS. Finally, we obtained the highest gain (0.28 dB) by the model with LRSAM, CRFS, and GGS simultaneously, which is the GPSR model.

**Table 3.** Investigations of LRSAM, CRFS, and GGS. We observe that the proposed GPSR, that is, the model that uses LRSAM, CRFS, and GGS simultaneously, achieves the best super-resolution performance on Set5 dataset with scaling factor ×4.

| LRSAM |  | ✓ |  |  | ✓ | ✓ |  | ✓ |
| CRFS |  |  | ✓ |  | ✓ |  | ✓ | ✓ |
| GGS |  |  |  | ✓ |  | ✓ | ✓ | ✓ |
| PSNR | 32.46 | 32.52 | 32.52 | 32.61 | 32.63 | 32.70 | 32.67 | 32.74 |

### 4.3. Results with Bicubic (BI) Degradation

We compare the proposed GPSR with the nine state-of-the-art methods, including LapSRN [38], HAN [12], DBPN [34], RDN [11], EDSR [10], SAN [15], PA-EDSR [14], DRN [40], and NLSN [13] in terms of quantitative results and visual quality. Following [10,29], we also propose a self-ensemble model and denote it as GPSR+.

#### 4.3.1. Quantitative Metrics

For quantitative comparison, we compare the PSNR and SSIM metrics of different methods for the SR task with scaling factors ×4 and ×8. Table 4 lists the comparison of ×4 and ×8 quantitative results on the five commonly benchmark datasets, including Set5, Set14, BSD100, Urban100, and Manga109, where the optimal and suboptimal metrics are highlighted in red and blue, respectively, and where '–' means that the result is not available. Params and FLOPs denote the total number of parameters and floating-point operations, respectively. Noted that each the efficiency proxy FLOPs is measured under the setting of upscaling reconstructed images to the size of $1280 \times 720$ on the scaling factors ×4 and ×8. It should be noted that we derive the results of the methods used for comparison from their pre-trained models, released code, or their original paper. It is worth noting that some comparison methods, including HAN, RDN, EDSR, PA-EDSR, and NLSN, do not offer experimental results for ×8 SR. Therefore, we retrained these models on the ×8 scaling factor with their official source codes. In addition, for a fair comparison, we retrained DRN on the DIV2K dataset, as it originally used the DF2K dataset as the training set.

For convenience to quantitative comparisons, we take Set5 (×4) as an example. From Table 4, we observe that SR models #1 with deeper network structures and more extensive connections, including DBPN, RDN, and EDSR, obtain approximately 0.9 dB of PSNR gains compared to LapSRN. Further, SR models #2 with attention mechanisms, including HAN, SAN, PA-EDSR, and NLSN, can adaptively emphasize useful feature information for more accurate image reconstruction. The PSNR gains of models #2 over #1 range from 0.12 dB to 0.19 dB. However, the above models ignore the potential of the gradient prior for image SR. In contrast, the proposed GPSR makes full use of the gradient map to restore sharp high-frequency details, and obtains the highest quantitative metrics. It is worth noting that the proposed GPSR achieves a performance advantage of 0.09 dB with fewer parameters (18% parameters of PA-EDSR) and FLOPs (34% Flops of PA-EDSR) compared to PA-EDSR, which obtains sub-optimal SR results.This indicates that the proposed GPSR takes into account the computational efficiency while pursuing performance.

From Table 4, we can see that GPSR shows superior performance for both ×4 and ×8 SR tasks. GPSR+ achieves the optimal results on all test datasets. Compared with the nine state-of-the-art methods, the PSNR and SSIM metrics obtained by our GPSR are optimal in most test datasets. Specifically, our GPSR performs optimally on all datasets for both ×4 and ×8 SR tasks, except for the suboptimal SSIM metric on BSD100 for the ×4 SR task. Consequently, Table 4 provides adequate evidence that the proposed GPSR achieves the optimal super-resolution performance in the SR tasks with large scaling factors. Furthermore, Table 4 also presents the number of parameters and FLOPs for different SR methods. Overall, the proposed GPSR achieves a better tradeoff between computational efficiency and the super-resolution performance.

We also present an illustrated comparison in terms of the PSNR metric and the number of parameters for both the ×4 and ×8 SR tasks in Figure 8. As can be seen from Figure 8, compared with other SR models used for comparison, the proposed GPSR achieves the highest PSNR with a small number of parameters on the Set5 dataset for both the ×4 and ×8 SR tasks, indicating its optimal super-resolution performance.

**Table 4.** Quantitative metrics obtained by various models for SR task with BI degradation model. The optimal and suboptimal metrics are highlighted in red and blue.

| Method | Params | FLOPs | Scale | Set5 PSNR/SSIM | Set14 PSNR/SSIM | BSD100 PSNR/SSIM | Urban100 PSNR/SSIM | Manga109 PSNR/SSIM |
|---|---|---|---|---|---|---|---|---|
| LapSRN [38] | 0.9M | - | | 31.54/0.8850 | 28.19/0.7720 | 27.32/0.7270 | 25.21/0.7560 | 29.09/0.8900 |
| HAN [12] | 16.07M | 944G | | 32.64/0.9002 | 28.90/0.7890 | 27.80/0.7442 | 26.85/0.8094 | 31.42/0.9177 |
| DBPN [34] | 10.4M | 1106G | | 32.47/0.8983 | 28.75/0.7859 | 27.67/0.7396 | 26.38/0.7947 | 30.90/0.9134 |
| RDN [11] | 22.42M | 1309G | | 32.47/0.8990 | 28.81/0.7871 | 27.72/0.7419 | 26.61/0.8028 | 31.00/0.9151 |
| EDSR [10] | 43.1M | 2894G | | 32.46/0.8968 | 28.80/0.7876 | 27.71/0.7420 | 26.64/0.8033 | 31.03/0.9149 |
| SAN [15] | 15.9M | 936G | 4 | 32.64/0.9003 | 28.92/0.7888 | 27.79/0.7436 | 26.79/0.8068 | 31.18/0.9169 |
| PA-EDSR [14] | 45.53M | 3214G | | 32.65/0.9006 | 28.87/0.7891 | 27.76/0.7445 | 27.01/0.8140 | 31.29/0.9194 |
| DRN [40] | 4.8M | 685G | | 32.61/0.8974 | 28.89/0.7876 | 27.74/0.7389 | 26.71/0.8038 | 31.33/0.9150 |
| NLSN [13] | 46.52M | 2955G | | 32.59/0.9000 | 28.87/0.7891 | 27.78/0.7444 | 26.96/0.8109 | 31.27/0.9184 |
| GPSR | 8.18M | 1098G | | 32.74/0.9010 | 28.98/0.7901 | 27.81/0.7441 | 26.96/0.8116 | 31.50/0.9200 |
| GPSR+ | 8.18M | - | | 32.82/0.9022 | 29.06/0.7913 | 27.86/0.7451 | 27.14/0.8150 | 31.80/0.9223 |
| LapSRN [38] | 1.3M | - | | 26.15/0.7380 | 24.35/0.6200 | 24.54/0.5860 | 21.81/0.5810 | 23.39/0.7350 |
| HAN [12] | 16.22M | 271G | | 27.33/0.7884 | 25.24/0.6510 | 24.98/0.6059 | 22.98/0.6437 | 25.20/0.8011 |
| DBPN [34] | 23.2M | 236G | | 27.21/0.7844 | 25.13/0.6489 | 24.88/0.6010 | 22.72/0.6315 | 25.50/0.7984 |
| RDN [11] | 22.42M | 327G | | 26.91/0.7717 | 24.92/0.6405 | 24.85/0.5993 | 22.52/0.6211 | 24.60/0.7768 |
| EDSR [10] | 45.5M | 1271G | | 27.22/0.7840 | 25.14/0.6476 | 24.88/0.6010 | 22.70/0.6314 | 24.85/0.7906 |
| SAN [15] | 16M | 269G | 8 | 27.22/0.7840 | 25.14/0.6476 | 24.88/0.6010 | 22.70/0.6314 | 24.85/0.7906 |
| PA-EDSR [14] | 45.53M | 872G | | 27.05/0.7796 | 25.05/0.6456 | 24.93/0.6034 | 22.85/0.6393 | 24.97/0.7915 |
| DRN [40] | 5.4M | 171G | | 27.27/0.7873 | 25.21/0.6500 | 24.96/0.6040 | 22.93/0.6385 | 25.21/0.8016 |
| NLSN [13] | 46.52M | 738G | | 26.57/0.7586 | 24.60/0.6299 | 24.66/0.5920 | 22.14/0.6011 | 24.00/0.7546 |
| GPSR | 14.42M | 382G | | 27.45/0.7935 | 25.29/0.6536 | 25.00/0.6059 | 23.08/0.6474 | 25.40/0.8078 |
| GPSR+ | 14.42M | - | | 27.52/0.7956 | 25.38/0.6555 | 25.05/0.6070 | 23.23/0.6518 | 25.64/0.8116 |

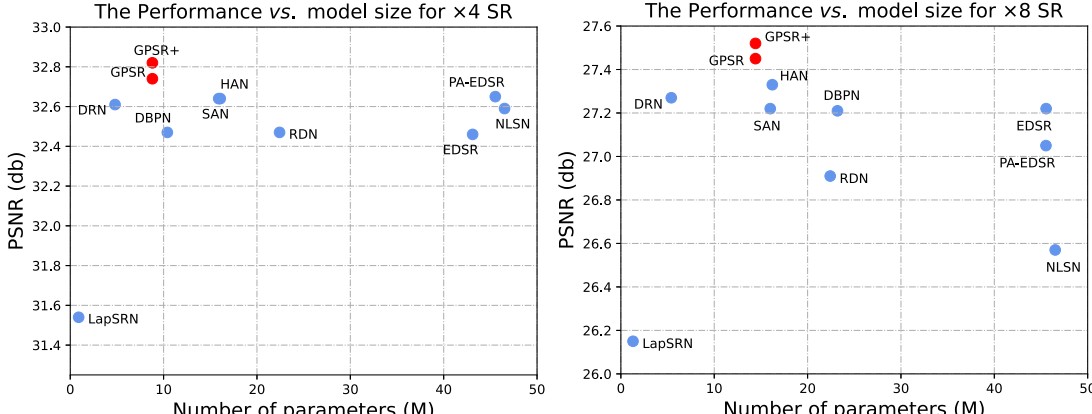

**Figure 8.** Comparisons of the PSNR metric and the number of parameters among different SR models on the Set5 dataset.

### 4.3.2. Visual Quality

Figure 9 illustrates the visual comparisons of ×4 and ×8 SR tasks, respectively, which offer the reconstructed results obtained using different methods in the same patches, and the corresponding original HR patches are given as references. From Figure 9, we can see that most of the methods used for comparison are weak in reconstructing sharp edge and texture details, and even produce severe geometric distortions and artificial artifacts. Compared with these methods, our GPSR is able to reconstruct a clearer image with sharp edge and texture details. Remarkably, GPSR suppresses the undesired geometric distortions well, and this advantage becomes increasingly important as the scaling factor increases in SR tasks.

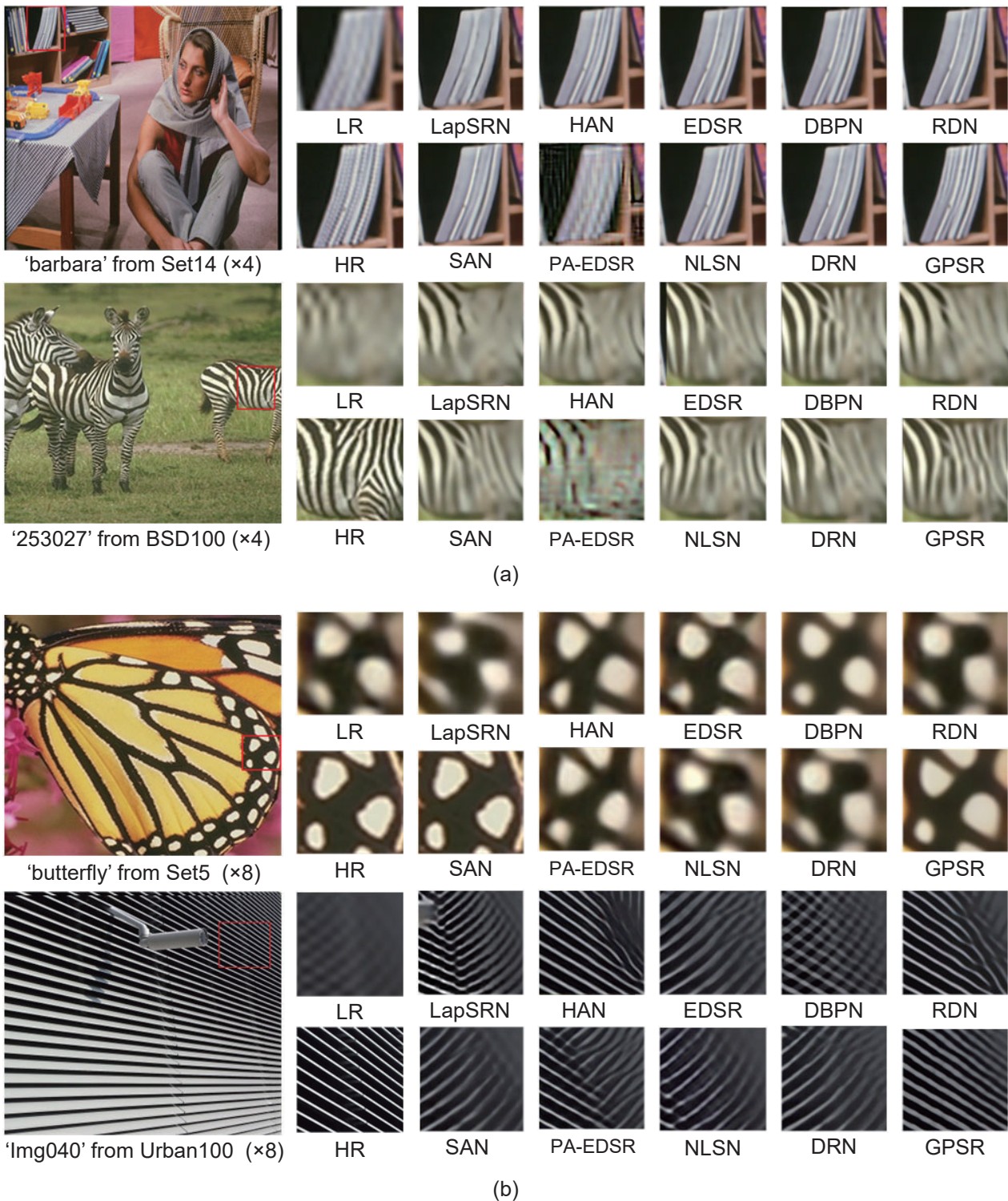

**Figure 9.** Visual comparisons for SR tasks. (**a**) Visual comparison for ×4 SR task with BI model on Set14 and BSD100 datasets. (**b**) Visual comparison for ×8 SR task with BI model on Set14 and Urban100 datasets.

In Figure 9a, we show visual comparisons of the ×4 SR task. For the image "barbara" from the Set14 dataset, we observe that most of the compared methods, such as LapSRN, EDSR, RDN, and DRN fail to recover the edge details of the book effectively, making their reconstructed images appear blurred and smooth. Compared with the previous methods, SAN and NLSN show better detail recovery ability, but their reconstructed images are still

unsatisfactory. Compared with the compared methods, the proposed GPSR shows better structure preservation and detail recovery performance, which is reflected in sharper and more complete edge details. From the image "253027" in the BSD100 dataset, the zebra stripes produced by most of the compared methods are blurred and confused, and there exist different degrees of information loss in the local regions. However, DRN, SAN, and NLSN retain more texture than LapSRN. However, they are still inferior to our GPSR. This is due to the fact that benefits from our GGS, the proposed GPSR, are able to effectively local gradient information to reconstruct high-frequency details while avoiding the problem of blindly restoring details that exist in most comparative models.

Figure 9b presents visual comparisons of the ×8 SR task. For the image "butterfly" from Set5, we observe that most of the compared methods fail to restore the spots and suffer from blurred artifacts. In contrast, our GPSR is able to alleviate the blurred artifacts better, and reconstruct sharp edge and texture details. For the image "img040" from Urban100, the reconstructed images produced by most comparative methods have significant geometric distortions, such as significant errors in the extension directions of the edges. Although HAN, RDN, and DBPN are able to suppress geometric distortions to a certain extent effectively, this deficiency still exists in local regions caused by blind detail restoration. The optimal reconstructed image is generated by the proposed GPSR, which again proves the effectiveness of the proposed GGS.

### *4.4. Results with Blur-Downscale (BD) Degradation*

Following [29], we also test the effectiveness of the proposed GPSR on the LR images with the blur-down (BD) degradation model. We selected the five state-of-the-art methods, including RCAN [29], DRN [40], HAN [12], PA-EDSR [14], and NLSN [11], to compare with our GPSR in objective quantitative metrics. Table 5 depicts the results of different SR models for the ×8 SR task. It can be seen from Table 5 that our GPSR also consistently outperforms other compared methods, even without self-ensemble in the SR task with large scaling factor. Specifically, the PSNR gain of our GPSR over NLSN is up to 1.05 dB and 1.25 dB on the Urban100 and Manga109 datasets, respectively.

**Table 5.** Quantitative metrics obtained using various models for ×8 SR task with BD degradation model. The optimal and suboptimal metrics are highlighted in red and blue.

| Method | Params | FLOPs | Scale | Set5 PSNR/SSIM | Set14 PSNR/SSIM | BSD100 PSNR/SSIM | Urban100 PSNR/SSIM | Manga109 PSNR/SSIM |
|---|---|---|---|---|---|---|---|---|
| DRN [40] | 5.4M | 171G | | 26.17/0.7704 | 24.43/0.6394 | 24.26/0.5916 | 22.22/0.6315 | 23.70/0.7782 |
| RCAN [29] | 15.74M | 264G | | 26.22/0.7713 | 24.51/0.6415 | 24.28/0.5933 | 22.34/0.6402 | 23.87/0.7806 |
| HAN [12] | 16.22M | 271G | | 26.20/0.7612 | 24.43/0.6346 | 24.28/0.5888 | 22.33/0.6286 | 23.71/0.7629 |
| PA-EDSR [14] | 45.53M | 1271G | 8 | 26.26/0.7727 | 24.51/0.6401 | 24.29/0.5935 | 22.57/0.6460 | 23.85/0.7780 |
| NLSN [11] | 46.52M | 738G | | 25.91/0.7544 | 24.04/0.6245 | 24.10/0.5840 | 21.80/0.6062 | 23.00/0.7465 |
| GPSR | 14.42M | 382G | | 26.32/0.7788 | 24.60/0.6464 | 24.36/0.5974 | 22.85/0.6607 | 24.25/0.7950 |
| GPSR+ | 14.42M | - | | 26.42/0.7809 | 24.72/0.6491 | 24.42/0.5987 | 23.05/0.6656 | 24.50/0.7993 |

## 5. Conclusions

In this paper, we fully explore the potential of the gradient prior for SR task. Specifically, to restore sharp high-frequency details, we introduce an additional gradient branch in the classical SR network to provide the beneficial structural features for each upsampling stage of the SR process. Meanwhile, we propose a compact Gradient-Guided (GG) loss to strengthen the constraints on the spatial structure of the reconstructed images, so as to guide the model to restore the appropriate gradient information at the suitable position and to avoid blindly restoring high-frequency details. Additionally, we propose a novel Contextual Residual Fusion Structure (CRFS), which is capable of fully fusing complementary contextual residual features to produce more advanced features; further, to promote the feature representation ability of CRFS as much as possible, we propose an efficient Large-Receptive-field-based Spatial Attention Module (LRSAM) to highlight the

critical residual features; finally, we incorporate LRSAM into CRFS and further propose two Residual Spatial Attention Adaptive Aggregation Modules (RS3AMs) used for image feature extraction and gradient feature extraction, respectively. Extensive experiments indicate that the superiority of our Gradient-Prior-based Super-Resolution network (GPSR) for images over state-of-the-art methods for SR tasks with large scaling factors ($\times 4$ and $\times 8$). However, there are two limitations to our method, as follows: (1) Although our GPSR achieves better results with fewer parameters and FLOPs than most existing SR methods, its computational efficiency still needs to be further improved to satisfy the requirements of real applications, such as the deployment of the technique in embedded devices. (2) Like most existing SR methods, the proposed GPSR only considers known degradation models (e.g., BI and BD degradation kernels), and it is still struggling with SR tasks with unknown degradation kernels. To address these two limitations, we will focus on compressing the model to improve computational efficiency while retaining SR performance in our future work. Moreover, we will explore the introduction of the adversarial generative network into the SR task to estimate the unknown degradation kernels, which facilitates the real-world image SR.

**Author Contributions:** X.Z.: Conceptualization, Methodology, and Writing. D.H.: Supervision, Review, and Editing. X.L.: Software and Visualization. D.C.: Original draft preparation. D.Z.: Editing and Review. All authors have read and agreed to the published version of the manuscript.

**Funding:** This work was supported in part by the National Natural Science Foundation of China under Grant 61901183, in part by the National Key R&D Program of China under the grant 2021YFE0205400, in part by the Collaborative Innovation Platform Project of Fujian Province under Grant 2021FX03, in part by the Natural Science Foundation of Fujian Provincial Science and Technology Department under Grant 2021H6037, in part by the Natural Science Foundation of Fujian Province under Grant 2019J01010561, in part by the Fundamental Research Funds for the Central Universities Grant ZQN-921, in part by the Foundation of Fujian Education Department under Grant JAT170053, and in part by the Key Project of Quanzhou Science and Technology Plan under Grant 2021C008R.

**Institutional Review Board Statement:** Not applicable.

**Informed Consent Statement:** Not applicable.

**Data Availability Statement:** https://pan.baidu.com/s/1hXuPoIHf8A4wm1ZzM7rBfQ?pwd=Z983 (accessed on 23 November 2022), password: Z983.

**Acknowledgments:** 

**Conflicts of Interest:** No potential conflict of interests were reported by the authors. The authors declare that they have no known competing financial interests or personal relationships that could have appeared to influence the work reported in this paper.

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
