# Peer review of "GPSR: Gradient-Prior-Based Network for Image Super-Resolution"

_applsci, doi:10.3390/app13020833_

Round 1

Reviewer 1 Report

The authors present an interesting proposal focused on the prior gradient for the SR task. In terms of scientific soundness, experimental design and result, the authors have done a good job. The issues as listed below:

- In figure 5 (line 250) just put the indicative letters (a and b) below the figure. And in the figure caption the description of each one. The same for the following figures.

- In lines 271-273 you mention the datasets used, make a table where you explain the characteristics of each dataset, for example, number of images, size, among others.

- On line 278, in YcbCr, the first c should also be capitalized. On the other hand, why is the image converted to YCbCr space? Add the importance of using this color space

- In line 282 explain the reasons (advantages) for using the ADAM optimization method and not another.

- Some sections of figure 1 are small. Please improve it. In the same way, Figures 3, 4 and 5.

- In the conclusions (line 452) you talk about computational efficiency, however, in the paper you do not mention the computational efficiency of your proposal. You can mention it in section 4.4, specifically in table 4, comparing with other methods.

- Although a comparison with other works is made, results need to be discussed.

- Authors need to provide the limitations of their experiments and techniques in detail in discussion or conclusion.

Reviewer 2 Report

Authors in this research proposed a new deep neural network for image super-resolution. In this network, a gradient branch is added to the commonly used super-resolution branch. A detail-preserving Gradient Guidance Strategy (GGS), a novel Contextual Residual Fusion Structure (CRFS), and a large-receptive-field-based spatial attention module (LRSAM) are used in the network. Specifically, LRSAM is incorporated into CRFS to construct two Residual Space Attention Adaptive Aggregation Modules: RS3AM-SR and RS3AM-G for the SR branch and the gradient branch, respectively. All these components contribute to the recovery of high-frequency features robustly.

This work can be improved from the following aspects.

1. A GAN-based network can be used for super-resolution.

Although a GAN-based network may be uneasy to train, its’ structure can be very straightforward, and it can be very powerful. Is training difficulty a strong enough reason for abandoning the GAN direction?

2. The proposed network is very complex.

In practice, modifying, debugging, and training a complex network is highly nontrivial. How do the authors justify the proposed method is easier to be used than a GAN-based network?

3. Are the comparisons with other methods fair?

In a large dataset, it is easy to find an image preferable for each method. In Figure 9, are the images chosen so that the proposed method appears to be the best choice?

4. Quantitative comparisons need to be improved.

The numbers in Table 3, Figure 8, and Table 4 are close. Standard deviations need to be added to the numbers and statistic tests need to be performed.

5. Down-sampled images need to be added to Figure 9.

Round 2

Reviewer 1 Report

The comments have been properly addressed. Thank you.